# Inter-Scale Dependency Modeling for Skin Lesion Segmentation with Transformer-based Networks

**Sania Eskandari**[*1]                                                       SES235@UKY.EDU
**Janet Lumpp**[1]                                                          JKLUMPP@UKY.EDU
[1] *Department of Electrical and Computer Engineering, University of Kentucky, Lexington, USA*

**Editors:** Accepted for publication at MIDL 2023

## Abstract

Melanoma is a dangerous form of skin cancer caused by the abnormal growth of skin cells. Fully Convolutional Network (FCN) approaches, including the U-Net architecture, can automatically segment skin lesions to aid diagnosis. The symmetrical U-Net model has shown outstanding results, but its use of a convolutional operation limits its ability to capture long-range dependencies, which are essential for accurate medical image segmentation. In addition, the U-shaped structure suffers from the semantic gaps between the encoder and decoder. In this study, we developed and evaluated a U-shaped hierarchical Transformer-based structure for skin lesion segmentation while we proposed an Inter-scale Context Fusion (ISCF) to utilize the attention correlations in each stage of the encoder to adaptively combine the contexts coming from each stage to hinder the semantic gaps. The preliminary results of the skin lesion segmentation benchmark endorse the applicability and efficacy of the ISCF module.

**Keywords:** Deep learning, Transformer, Skin lesion segmentation, Inter-scale context fusion

## 1. Introduction

Automatic segmentation of organs is an essential cue for developing the pre and post-diagnosis process with computer-aided diagnosis (CAD), while manual delineation is a tedious and laborious task. Skin cancer is a dangerous and often deadly disease. The skin comprises three layers: the epidermis, dermis, and hypodermis. When exposed to ultraviolet radiation from the sun, the epidermis produces melanin, which can be produced at an abnormal rate if too many melanocytes are present. Malignant melanoma is a deadly form of skin cancer caused by the abnormal growth of melanocytes in the epidermis, with a mortality rate of 1.62%. In 2022, it was estimated that there would be 99,780 new cases of melanoma with a mortality rate of 7.66% (Siegel et al., 2022). The survival rate drops from 99% to 25% when melanoma is diagnosed at an advanced stage due to its aggressive nature. Therefore, early diagnosis is crucial in reducing the number of deaths from this disease. Utilizing U-Net, with a hierarchical encoder-decoder design in semantic segmentation tasks, is a common choice. The U-shaped structure leverages some advantages, making it an ideal choice for skin lesion segmentation tasks. However, the conventional design suffers from limited receptive fields due to the convolution operations' presence in the framework. Therefore, Vision Transformer (ViT), as a drift from the natural language processing's eminent counterpart, Transformers, adapted to the wide range of vision architectures to capture

---

* Corresponding author

long-range dependencies. While the computational complexity of ViTs is proportional to the number of patches and quadratic, utilizing the standalone ViT as a main backbone for dense prediction tasks, *e.g.,* segmentation, is problematic. On the other hand, due to the need to design a hierarchical pipeline for segmentation tasks, ViTs, in their conventional aspect, is not desirable. Thus, various studies explored minimizing this computational burden to make ViTs ready to participate in segmentation tasks by delving into the inner structure of the Transformer's multi-head self-attention (MHSA) calculation or changing the tokenization process such as the Efficient Transformer (Xie et al., 2021), and the Swin Transformer (Liu et al., 2021). Moreover, an analytic demonstration of MHSA by (Wang

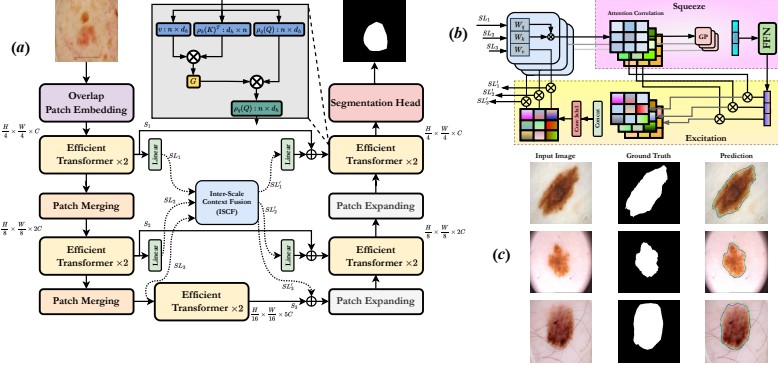

Figure 1: (**a**) The overall end-to-end proposed pipeline for skin lesion segmentation with Efficient Transformer (Shen et al., 2021) in a U-shaped structure. (**b**) Inter-Scale Context Fusion Module. (**c**) Qualitative results on *ISIC 2018* dataset. Green represents the ground truth contour and blue denotes the prediction mask contour.

et al., 2022) revealed that the Transformers perform as a low-pass filter due to the **Softmax** non-linear operation. This deficiency further degrades the Transformer's applicability for dense semantic segmentation tasks, while the conventional U-shaped methods already need to improve from the semantic gaps between the encoder and decoder. Thus, we alleviate the losing high-frequency input counterparts in stacked Transformer-based structures besides hindering the semantic gap between the encoder and decoder in an adaptive technique.

Contrary to the Swin Transformer (Liu et al., 2021), we utilize the Efficient Transformer block to hinder the loss of contextual information in the windowing strategy for MHSA. One significant drawback is MHSA can extract the limited context within windows. In a window-based Transformer, the input sequence is divided into fixed-size windows, and self-attention is only applied within each window. This means long-range dependencies between elements in different windows may not be effectively captured. This can limit the ability of window-based Transformers to model complex patterns and relationships in sequential data. In this paper, we used the Efficient Transformer (Shen et al., 2021) as a main block in our U-shaped Transformer-based pipeline as in Figure 1(a) to compensate for the Swin Transformer's mentioned deficiency. U-shaped structures suffer from the semantic gap between the encoder and decoder. To this end, we were inspired by the squeeze and excitation paradigm and proposed a **I**nter-**S**cale **C**ontext **F**usion (**ISCF**) to alleviate the mentioned semantic gap (see Figure 1(b)).

## 2. Method

Our proposed U-shaped structure is defined as three stages multi-scale manner coupled with an ISCF module. Due to the hierarchical design of the structure, the attention maps' shape at each level differs from the next one. Therefore, we used a Linear layer in the first two stages two make the attention map sizes as same as in the last stage. This operation is done at the output of the ISCF module to remap the attention maps to their original sizes. In the ISCF module, we utilize the Global Pooling (GP) operation to produce a single value for each stage's attention correlation and concatenate them, followed by a Feed Forward Network (FFN) to amalgamate the contribution of each global value with each other as a scaling factor. Then each attention map applies the Hadamard production with the corresponding scaling value and concatenates the resultant attention maps. Finally, to adaptively amalgamate these global contexts with each other to lessen the mentioned semantic gaps, a $3 \times 1 \times 1$ is used. We use the publicly available *ISIC 2018* skin lesion benchmark dataset that contains 2,594 images for the evaluation process. We resized each sample to $224 \times 224$ pixels from $576 \times 767$ and used 1,815 samples for training, 259 samples for validation, and 520 samples for testing. Our proposed method is implemented end-to-end using the PyTorch library and is trained on a single Nvidia RTX 3090 GPU. The training is done with a batch size of 24 and an Adam optimizer with a learning rate of 1e-4, which was carried out for 100 epochs, and for the loss function, we used binary cross-entropy.

## 3. Results

In Table 1, the quantitative results for our proposed method are displayed. We reported the performance of the model on the Dice score (DSC), sensitivity (SE), specificity (SP), and accuracy (ACC). The preliminary results show that our design can outperform SOTA methods without pre-training weights and having fewer parameters. In addition, Figure 1(c) represents the qualitative results that the network performs well with respect to the ground truth results and preserves the high-frequency details such as boundary information.

Table 1: Performance comparison on the *ISIC 2018* skin lesion segmentation dataset.

| Methods | # Params(M) | DSC | SE | SP | ACC |
|---|---|---|---|---|---|
| U-Net (Ronneberger et al., 2015) | 14.8 | 0.8545 | 0.8800 | 0.9697 | 0.9404 |
| Att U-Net (Oktay et al., 2018) | 34.88 | 0.8566 | 0.8674 | **0.9863** | 0.9376 |
| TransUNet (Chen et al., 2021) | 105.28 | 0.8499 | 0.8578 | 0.9653 | 0.9452 |
| FAT-Net (Wu et al., 2022) | 28.75 | 0.8903 | 0.9100 | 0.9699 | 0.9578 |
| Swin U-Net (Cao et al., 2023) | 82.3 | 0.8946 | 0.9056 | 0.9798 | **0.9645** |
| Efficient Transformer (without ISCF) | 22.31 | 0.8817 | 0.8534 | 0.9698 | 0.9519 |
| **Efficient Transformer (with ISCF)** | **23.43** | **0.9136** | **0.9284** | 0.9723 | 0.9630 |

## 4. Conclusion

The semantic gap between the encoder and decoder in a U-shaped Transformer-based network can be mitigated by carefully recalibrating the already calculated attention maps from each stage. In this study, not only do we address the hierarchical semantic gap drawback, but also we compensate for the deep Transformers' high-frequency losses by utilizing the earlier Transformer's attention map by the ISCF. ISCF module is a plug-and-play and computation-friendly module that can effectively be applied to any Transformer-based architecture.

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
