# OpenReview forum: "Inter-Scale Dependency Modeling for Skin Lesion Segmentation with Transformer-based Networks"
_MIDL.io/2023/Short_Paper_Track — MIDL 2023 Short paper track Poster_

### Official Review · Reviewer_44ZD · 2023-04-19
**Well validated but good be better explained**

**Rating:** 7
**Confidence:** 5

**Review:**

This paper proposes use of an efficient transformer architecture to overcome some of the limitations of the computationally inefficient ViT and approximate Swin transformer, for the task of skin lesion segmentation. The results seem well validated, but the efficient transformer isn’t well explained so it’s difficult to get a handle on why it presents an effective solution to the problem. Swin transformer isn’t cited.

---

### Official Review · Reviewer_QSuG · 2023-04-23
**This paper presents a skin lesion segmentation model using a a hierarchical Transformer-based structure with an Inter-scale Context Fusion (ISCF) module to address the limitations of convolutional operations and semantic gaps between the encoder and decoder. The preliminary results show that the proposed ISCF module is effective in improving skin lesion segmentation accuracy based on a benchmark evaluation.**

**Rating:** 6
**Confidence:** 4

**Review:**

**Pros:**

- The study incorporates contextual information in a transformer-based segmentation model for medical imaging, which can enhance global context understanding.
- The proposed method is evaluated on a publicly available benchmark dataset with satisfactory results, demonstrating its applicability and efficacy.
- The Inter-scale Context Fusion (ISCF) module improved model performance while keeping trainable parameters low, similar to a squeeze and excitation block, which can be beneficial for practical implementation.
- The Efficient Transformer with ISCF outperforms other segmentation models in the context of skin lesion segmentation.

**Cons:**

- The generalizability of the proposed approach to other segmentation tasks is a concern, and further testing on diverse datasets is needed to quantify its effectiveness in different medical imaging scenarios, such as multi-organ segmentation where contextual information is critical due to overlapping organs.
- It is unclear if Swin-UNet is the same as SwinUNETR (Hatamizadeh et al. 2022) and proper citation and clarification are needed.
- More visual analysis is required to justify the impact of ISCF compared to Swin transformer.
- The introduction section needs proper citation to reference methods to ensure academic integrity.